# Analysis of Stored Wheat Grain-Associated Microbiota Reveals Biocontrol Activity among Microorganisms against Mycotoxigenic Fungi

**DOI:** 10.3390/jof7090781

**Published:** 2021-09-20

**Authors:** Manoj Kumar Solanki, Ahmed Abdelfattah, Sudharsan Sadhasivam, Varda Zakin, Michael Wisniewski, Samir Droby, Edward Sionov

**Affiliations:** 1Volcani Center, Agricultural Research Organization, Institute of Postharvest and Food Sciences, Rishon LeZion 7528809, Israel; mkswings321@gmail.com (M.K.S.); sudharsan@volcani.agri.gov.il (S.S.); veredz@volcani.agri.gov.il (V.Z.); 2Institute of Environmental Biotechnology, Graz University of Technology, Petersgasse 12, A-8010 Graz, Austria; ahmed.abdelfattaah@gmail.com; 3Agricultural Research Service, United States Department of Agriculture (USDA-ARS), 2217 Wiltshire Road, Kearneysville, WV 25430, USA; Michael.Wisniewski@ars.usda.gov

**Keywords:** biocontrol, stored wheat grain microbiota, epiphytes, endophytes, mycotoxigenic fungi, mycotoxins

## Abstract

Wheat grains are colonized by complex microbial communities that have the potential to affect seed quality and susceptibility to disease. Some of the beneficial microbes in these communities have been shown to protect plants against pathogens through antagonism. We evaluated the role of the microbiome in seed health: in particular, against mycotoxin-producing fungi. Amplicon sequencing was used to characterize the seed microbiome and determine if epiphytes and endophytes differ in their fungal and bacterial diversity and community composition. We then isolated culturable fungal and bacterial species and evaluated their antagonistic activity against mycotoxigenic fungi. The most prevalent taxa were found to be shared between the epiphytic and endophytic microbiota of stored wheat seeds. Among the isolated bacteria, *Bacillus* strains exhibited strong antagonistic properties against fungal pathogens with noteworthy fungal load reduction in wheat grain samples of up to a 3.59 log_10_ CFU/g compared to untreated controls. We also found that a strain of the yeast, *Rhodotorula glutinis*, isolated from wheat grains, degrades and/or metabolizes aflatoxin B_1_, one of the most dangerous mycotoxins that negatively affects physiological processes in animals and humans. The mycotoxin level in grain samples was significantly reduced up to 65% in the presence of the yeast strain, compared to the untreated control. Our study demonstrates that stored wheat grains are a rich source of bacterial and yeast antagonists with strong inhibitory and biodegradation potential against mycotoxigenic fungi and the mycotoxins they produce, respectively. Utilization of these antagonistic microorganisms may help reduce fungal and mycotoxin contamination, and potentially replace traditionally used synthetic chemicals.

## 1. Introduction

Wheat is one of the major cultivated grain crops, and an important source of calories and plant-derived protein in the human diet. The maintenance of high-quality and healthy wheat grains is essential to ensuring the stability of the world’s food supply and providing global food security. Wheat grains, like seeds of other cereal crops, are colonized by complex epiphytic and endophytic microbial communities that play an important role in grain health, quality, and susceptibility to disease [1]. Microorganisms, both bacteria and fungi, naturally occur in cereal crops without causing any damage, and may influence host growth and development. Conversely, other microorganisms can cause disease and spoilage, thereby decreasing crop value, and may also have a harmful effect on human health. For example, many species of *Fusarium*, *Aspergillus*, *Penicillium*, and *Alternaria* are not only recognized as plant pathogens but also as sources of important mycotoxins of concern to animal and human health [2,3]. Poor postharvest management can lead to rapid deterioration in grain quality, with severe decreases in germinability and nutritional value of the stored grain, possibly accompanied by undesirable fungal contamination and, consequently, toxin production [4]. Mycotoxigenic fungal species may also cause significant wheat grain yield losses in field and storage facilities due to their ability to produce mycotoxins and render the crops unsafe for consumption [5,6,7,8,9]. Despite the major threat posed by these fungi, effective control measures are lacking. Current disease-management practices are primarily based on the use of synthetic fungicides, which have proven to be effective against several genera of mycotoxigenic fungi, applying them in the field before harvest. Increasing awareness of the possible harmful effects of these chemical compounds on the environment and human health, however, has fostered research to find effective and less toxic alternative strategies to control fungal infections. 

Biological control using microbial antagonists against fungal plant pathogens is one alternative approach that can be used as part of an integrated management strategy for the control of mycotoxigenic fungi and mycotoxin contamination in stored wheat. The protective ability of some bacterial epiphytes and endophytes against fungal pathogens has been reported in several crops, including wheat [1,10,11,12,13]. Nevertheless, one of the major drawbacks associated with the use of biological control agents is their inconsistent efficacy over time [14,15]. This can be attributed in part to the fact that most commercially available biocontrol isolates do not originate from the plants that they are intended to protect, a factor that may be related to their inconsistent performance in the field and/or storage warehouses. Exploring the epiphytic and endophytic composition of the stored wheat grain microbiome may help in identifying novel antagonistic microorganisms with potential antifungal and antitoxigenic activity. The major objective of the present study was to use amplicon sequencing to identify and characterize the epiphytic and endophytic microbiome of stored wheat grain. We then examined the potential of the culturable microorganisms isolated from stored seeds to inhibit the growth of mycotoxigenic fungi and degrade their toxins.

## 2. Materials and Methods

### 2.1. Wheat Grain Sources

Twenty-seven samples of wheat grain (*Triticum aestivum*) were collected from three wheat grain storage facilities located in northern and southern districts of Israel. In each of the three storage facilities, nine samples (1 kg of grain each), destined for human consumption, were collected in equal proportions (9 samples × 3 storage sites = a total of 27 samples) from the front face and center, at points located at 1 m horizontal depth within the grain mass, and from areas close to the walls. Grain temperature and moisture content were in the ranges of 27–33 °C and 10.5–12.9%, respectively. The collected samples were kept in sterile plastic bags during transport to the laboratory and stored at 4 °C (for up to 4 weeks) for further analysis. 

### 2.2. DNA Extraction, Amplification, and Sequencing

To isolate DNA from epiphytic microbiota, 10 g of each grain sample was soaked in 45 mL peptone water (10 g/L peptone, 5 g/L NaCl) (Difco; Becton Dickinson, Franklin Lakes, NJ, USA) containing 0.05% Triton X-100 (Sigma, Saint Louis, MO, USA) in a 250-mL Erlenmeyer flask and shaken at room temperature on an orbital shaker at 150 rpm for 1 h. Grains were aseptically removed by filtration through 4 layers of sterile cheesecloth and the remaining liquid fractions were centrifuged at 4000× *g* for 15 min. Pellets were collected and immediately frozen in liquid nitrogen, freeze-dried, then subjected to DNA extraction. DNA from endophytic microbiota was isolated after surface sterilization of wheat grains with 3% sodium hypochlorite for 2 min, followed by 70% ethanol for 2 min and rinsing five times in sterile distilled water. Surface-sterilized wheat grains (10 g) were frozen in liquid nitrogen, freeze-dried and milled into a fine powder with a grain grinder. The grain grinder was cleaned and disinfected with 70% ethanol solution between samples and the powder was used for DNA isolation. Extraction of both epiphytic and endophytic DNA samples was performed using a previously described protocol [16]. The purity of the extracted DNA was assayed with a Nanodrop™ One spectrophotometer (Thermo Scientific, Wilmington, DE, USA), and the total DNA concentration in each sample was adjusted to 50 ng/μl. The universal primers 799F/1392R [17] and 5F/86R [18] were used to amplify the 16S and ITS2 rRNA gene regions of bacteria and fungi, respectively (Appendix A). The primers were modified to include Illumina adapters (www.illumina.com (accessed on 7 March 2019)) for subsequent multiplexing. PCR amplification of each sample was performed in triplicate. The PCR mixture (25 µL) contained 12.5 µL 2× DreamTaq Green PCR Master Mix (Thermo Scientific, Vilnius, Lithuania), 1 µL of each primer (5 µM), and 1 µL DNA template. Nuclease-free water (Thermo Scientific) replaced the DNA template in negative controls. All amplicons and amplification mixtures, including negative controls, were sequenced using Illumina MiSeq V3 (2 × 300 bp) chemistry.

### 2.3. Bioinformatics

Illumina adaptors were clipped and low-quality reads removed by Trimmomatic 0.36 [19] using sliding-window trimming, cutting once the average quality in a window of 4 bases fell below the quality threshold of 15. Paired-end reads were merged utilizing PEAR [20] for the 16S rRNA gene, and PANDAseq [21] for the ITS rRNA gene region sequences with default parameters. Chimeric sequences were identified and removed using USEARCH [22,23] for the 16S rRNA gene, and VSEARCH 1.4.0 [24] for ITS rRNA gene region sequences. UCLUST algorithm [22], as implemented in QIIME 1.9.1 [25], was used to cluster sequences queried against the Greengenes 13_8_97 database for 16S rRNA genes [26], and for ITS UNITE dynamic database released on 1 December 2017 [27] at a similarity threshold of 97%, respectively. Sequences that failed to cluster against the database were de novo clustered using the same algorithm. After removing singletons, the most abundant sequences in each operational taxonomic unit (OTU) were selected as representative sequences and used for the taxonomic assignment with the BLAST algorithm [23,28] as implemented in QIIME 1.9.1. The OTU table was normalized by rarefaction to an even sequencing depth in order to eliminate sample heterogeneity. The OTU tables were rarefied to 300 bacterial and 8000 fungal sequences per sample and used to calculate α-diversity indices, including observed species (S_obs_) and Shannon index. The α-diversities were compared based on a two-sample t-test using nonparametric (Monte Carlo) methods and 999 Monte Carlo permutations. Results were visualized in boxplot figures. 

MetagenomeSeq’s Cumulative Sum Scaling (CSS) [29] was used as a normalization method for other downstream analyses. The CSS-normalized OTU table was analyzed using Bray–Curtis metrics [30] and utilized to evaluate β-diversity and construct principal coordinates analysis (PCoA) plots using Emperor [31]. The similarity in community composition was tested via ANOSIM in QIIME 1.9.1 using 999 permutations. Differential OTU abundance of the most abundant taxa (≥0.1%) between sample groups was determined by *t*-test and Kruskal–Wallis test [32]. In all tests, significance was determined using 999 Monte Carlo permutations; the false discovery rate (FDR) was used to adjust the calculated *p*-values and when the FDR *p* < 0.05, it was considered significant (Appendix A).

### 2.4. Isolation and Identification of Microorganisms

To isolate epiphytic microorganisms from wheat grains, 10 g of whole grain samples were incubated in 45 mL of sterile peptone water (Difco; Becton Dickinson, Franklin Lakes, NJ, USA) and shaken (150 rpm) at room temperature for 1 h. Then, 100-µL aliquots of serial dilutions of the broth were plated on Luria–Bertani (LB) agar plates for isolation of bacteria, and potato dextrose agar (PDA) plates supplemented with chloramphenicol (20 µg/ml) for isolation of filamentous fungi and yeasts. To isolate endophytic microbes, 5 g of sterilized, ground wheat grain samples were added to 45 mL of sterile peptone water, and 100-µL aliquots of serial dilutions were plated on LB agar plates for isolation of bacteria and PDA plates supplemented with chloramphenicol for isolation of filamentous fungi and yeasts. Inoculated LB agar plates were incubated at 37 °C for 24–48 h, whereas PDA plates were incubated at 28 °C for 48–72 h. Bacterial colonies were randomly selected from the plates and streaked on fresh culture media to obtain pure cultures. Filamentous fungi and yeasts were transferred singly to PDA plates and subcultured twice to obtain pure cultures. DNA was extracted from each bacterial strain using the lysozyme lysis method described by De et al. [33]. Fungal DNA extraction was performed on lyophilized mycelium/yeast cells using the CTAB-based method as previously described [16]. DNA quality and yield were determined using the Nanodrop One spectrophotometer. The 16S rRNA gene in bacteria, ITS rRNA gene region in fungi, and D1/D2 domain of large-subunit ribosomal DNA (rDNA) in yeast were amplified with universal primers 27F/1492R, ITS1/ITS4, and NL1/NL4, respectively (Appendix A). PCR products were purified and sequenced by standard Sanger sequencing; sequences were identified via BLAST matches to the NCBI database (https://blast.ncbi.nlm.nih.gov/Blast.cgi (accessed on 26 June 2021)) and deposited as accession numbers MZ452446–MZ452604 (bacterial 16S rRNA gene), MZ578164–MZ578238 (fungal ITS rRNA gene region) and MZ452263–MZ452328 (D1/D2 domain of large-subunit rDNA in yeast).

### 2.5. In Vitro Antagonism against Mycotoxigenic Fungi

Mycotoxigenic fungal isolates *Aspergillus flavus* (NRRL3518) and *Fusarium proliferatum* (NRRL31866) were obtained from USDA Agricultural Research Service Culture Collection (Northern Regional Research Laboratory, Peoria, IL, USA); *Alternaria infectoria* (strain F11) was isolated from stored wheat grains. Strains were refreshed from −80 °C by subculturing on PDA plates and maintained at 28 °C before each experiment. Conidia were collected in sterile saline and the conidial suspension was adjusted to the required concentration by counting in a hemocytometer. The inoculum of the test strains was verified by plating on PDA plates for determination of colony forming units (CFU) counts. The possible antagonistic effects of the obtained bacterial and yeast isolates against the mycotoxigenic fungi were assessed by dual-culture method. Each bacterial/yeast strain and fungal isolate were inoculated on a 90-mm PDA plate. Each tested bacterium or yeast was precultured overnight in LB broth or potato dextrose broth (PDB), respectively, and streaked in a straight line in the center of the 90-mm Petri dish, then 5 µL of fungal spore suspension at 10^7^ spores/mL was spotted at opposite edges of the plate. The plates were sealed with parafilm and incubated at 28 °C until the mycelia in the untreated controls reached the center of the plates. 

The production of volatile metabolites with antifungal activity released by bacterial and yeast isolates was also investigated. Precultivated overnight cultures of each bacterial or yeast strain were inoculated on a 55-mm LB or PDA agar plate, respectively, and incubated at 28 °C for 3 days. Then, the lids of these plates were replaced by the bottom of a PDA plate inoculated with a fresh fungal mycelial plug. Plates were sealed together with sticky tape to minimize gas exchange and further incubated for 5 days. Controls were prepared in a similar manner, but the bottom plate contained no bacteria/yeast. In all of these experiments, growth inhibition (mm) of the tested fungi was measured with a ruler and compared with the fungal growth in the untreated control plates. At least three independent repetitions of each test were conducted.

### 2.6. Biocontrol Activity of Selected Bacterial Strains and Yeast against Mycotoxigenic Fungi on Wheat Grains

Surface-disinfected seeds of wheat (10 g) were placed in sterile Petri dishes and inoculated with 1 mL of a cell suspension of overnight culture (OD_600_ = 1.00) of each bacterial and yeast isolate selected based on their inhibitory effect on fungal growth. Grains treated with sterile saline instead of bacterial/yeast inoculum were used as negative controls. After 24 h incubation at 28 °C, wheat grains were inoculated with 1 mL of 10^5^ conidia/mL of each fungal isolate. Untreated wheat grains were also inoculated with fungal conidia and served as positive controls. Plates were incubated at 28 °C for 7 days and fungal development was assessed visually. For each bacterial and/or yeast treatment of wheat grains inoculated with each fungal isolate, at least three independent experiments were conducted. The antifungal activity of bacterial and/or yeast strains on inoculated wheat grains was evaluated using the plate counting method and expressed as fungal colony-forming units (CFUs)/g dry weight of wheat grains. Briefly, grain samples (5 g) were suspended in 50 mL phosphate-buffered saline containing 0.05% Triton X-100 and incubated with shaking (160 rpm) at 28 °C for 30 min, then 100 µL of serial dilutions of the suspension were plated on PDA plates supplemented with chloramphenicol, followed by incubation at 28°C for 4 days, for counting of fungal CFUs.

### 2.7. Aflatoxin-Degradation Assays

Commercial aflatoxin B_1_ (AFB_1_) standard was purchased from Fermentek (Jerusalem, Israel). AFB_1_ at a concentration of 1 mg/mL was dissolved in methanol and stored at −20 °C. Based on previous reports indicating that the yeast *Rhodotorula glutinis* negatively affects mycotoxin accumulation in stored pome fruit [34], we included *R. glutinis* strain TY1, which we isolated from the wheat grains, in our mycotoxin-degradation assay. The analysis of AFB_1_ persistence in PDB medium was performed as follows. *R. glutinis* TY1 was grown overnight in 50 mL PDB with shaking at 28 °C. The culture was centrifuged for 5 min at 8000× *g*, the cells were resuspended in PDB, and their concentration was adjusted to 1 × 10^8^ CFU/ml. A 20-mL aliquot of this cell suspension was incubated on an orbital shaker for 3 days at 28 °C in a 100-mL flask in the presence of 250 ng/mL of AFB_1_. Controls included PDB medium only with 250 ng/mL of AFB_1_ and PDB without AFB_1_ inoculated with *R. glutinis* TY1 at the same cell concentration. Yeast growth was monitored on a daily basis by recording OD_600_ in the Nanodrop One spectrophotometer. At the same time points, the time course of AFB_1_ degradation was analyzed by high-performance liquid chromatography (HPLC).

AFB_1_ degradation by the *R. glutinis* TY1 isolate was also examined in stored wheat grains. Briefly, 10 g of surface-disinfected wheat grains were contaminated with AFB_1_ (final concentration 250 ng/g) and inoculated with strain TY1 (final concentration 1 × 10^8^ cells/g) on Petri dishes. Controls consisted of noninoculated wheat grains with and without AFB_1_ and TY1-inoculated grains without AFB_1_. The plates were prepared in triplicate and kept at 28 °C for up to 3 days. Then, the grain samples were freeze-dried and milled to a fine powder with a grain grinder. AFB_1_ was extracted after 24, 48, and 72 h and samples were analyzed quantitatively by HPLC. Experiments were performed three times. 

### 2.8. HPLC Analysis

For AFB_1_ extraction from PDB broth, the samples were centrifuged at 8000× *g* for 5 min to pellet out the cells, and 2 mL of the supernatant was mixed with 2 mL of chloroform and vortexed for 15 min. After discarding the upper phase, the chloroform phase was dried at 50 °C under a stream of nitrogen gas. For the mycotoxin analysis from wheat grains, 2.5 g of the ground seeds were mixed with 10 mL of 84:16 (*v/v*) acetonitrile:water and placed in an orbital shaker at 200 rpm for 1 h at room temperature. After centrifugation at 8500× *g* for 15 min, 2 mL of the supernatant was passed through a Supel^TM^ TOX AflaZea column (Supelco, Bellefonte, PA, USA); then, the collected sample was evaporated by nitrogen stream at 50 °C. 

The dried samples were reconstituted in 900 µL of 10% acetonitrile and 100 µL of trifluoroacetic acid solution (70% water, 20% trifluoroacetic acid and 10% acetic acid (*v/v*)) for 15 min at 50 °C. After incubation, the samples were filtered through a 0.22-µm PTFE membrane filter, and quantitatively analyzed by injection of 30 µL into a reversed phase HPLC/UHPLC system (Waters ACQUITY Arc, Milford, MA, USA) with gradient elution of 70% water, 15% acetonitrile and 15% methanol at 1 ml/min through a Kinetex 3.5 µm XB-C_18_ (150 × 4.1 mm) column (Phenomenex, USA). The column temperature was maintained at 35 °C. Three noncontaminated PDB and/or wheat grain samples were spiked with AFB_1_ standard solution at three concentrations to construct calibration curves, which were used for mycotoxin quantification. The AFB_1_ peak was detected with a fluorescence detector (excitation at 365 nm and emission at 455 nm) and quantified by comparing with calibration curves of the mycotoxin standard.

### 2.9. Statistical Analysis

Data presented in the figures are mean values of three independent experiments. Standard errors of the mean (SEM) are presented. Differences among treatments were analyzed by one-way analysis of variance followed by Duncan’s multiple range test to determine treatment effects, as well as significant differences between the strains (SPSS 16.0 software). AFB_1_ persistence in the biodegradation assays was analyzed by one-way analysis of covariance to determine statistically significant differences between treatment effects on mycotoxin concentration controlling for time (covariance).

## 3. Results and Discussion

### 3.1. Composition of Epiphytic and Endophytic Microbiota Associated with Stored Wheat Grain

After quality filtering and removing chimeric, chloroplast and mitochondrial sequences, 682,478 16S and 1,044,499 ITS sequences were retained and assigned to 63,108 and 114,849 bacterial and fungal OTUs, respectively.

The taxonomic assignment of 63,108 bacterial OTUs revealed 22 phyla in the epiphytic and endophytic datasets, including *Proteobacteria,* the predominant in the datasets (48.1% and 67.8%), followed by *Actinobacteria* (10.4% and 18.8%), *Firmicutes* (13.7% and 10.6%), *Bacteroidetes* (9.2% and 2.1%) and *Acidobacteria* (0.2% and 0.1%), respectively (Figure 1A). *Planctomycetes* (7.7%) and *Chloroflexi* (2.7%) were found only as endophytes. The relative abundance of the epiphytic and endophytic bacterial microbiota varied at the genus level. The most abundant endophytic bacterial genera were *Pantoea* (7.5%), *Ohtaekwangia* (5.8%), *Bacillus* (4.9%), *Burkholderia* (4.0%), *Actinobacteria* (3.9%), *Enterobacter* (3.7%), *Pseudomonas* (3.1%), *Caulobacter* (2.6%), *Staphylococcus* (2.5%), *Sphingomonas* (1.9%), *Wolbachia* (1.8%), *Klebsiella* (1.6%), *Micrococcus* (1.5%), *Stenotrophomonas* (1.3%), *Massilia* (1.1%), *Paenibacillus* (1.0%), and *Ralstonia* (1.0%) (Figure 1B). In the epiphytic bacterial community, the most abundant genera were *Pantoea* (20.3%), *Sphingomonas* (10.2%), *Pseudomonas* (8.1%), *Massilia* (8.5%), *Curtobacterium* (4.3%), *Microbacter* (4.3%), *Paracoccus* (3.3%), *Agrococcus* (2.7%), *Enterobacter* (2.7%), *Paenibacillus* (2.1%), *Enhydrobacter* (2.0%), *Hymenobacter* (1.4%), *Anaerococcus* (1.4%), *Acinetobacter* (1.2%), *Brevundimonas* (1.1%), and *Bacillus* (0.8%) (Figure 1B). These findings support earlier studies on seed-associated microbiota, where the bacterial taxa *Pantoea*, *Bacillus*, *Enterobacter,* and *Pseudomonas* were detected as both endophytic and epiphytic microorganisms [12,13,35,36].

Among all of the identified bacterial taxa, *Ruminococcaceae*, *SG8-4*, *Anaerolineaceae*, *Devosia*, *Actinobacteria*, *Diplorickettsiaceae*, *Pseudoalteromonas*, *Ohtaekwangia*, *Comamonas*, and *Klebsiella* were significantly (*p* < 0.05) higher in the wheat grain endosphere than in epiphytic communities. In contrast, bacterial epiphytic taxa such as *Microbacteriaceae*, *Sphingomonas*, *Agrococcus*, *Curtobacterium*, *Massilia*, *Hymenobacter*, *Methylobacterium*, *Paracoccus*, *Pseudomonas*, *Clavibacter*, *Pantoea, Enterobacteriaceae,* and *Arthrobacter* were significantly higher (*p* < 0.05) as epiphytes than as endophytes (Figure 1B). The differentially abundant or unique taxa characterizing the endophytic niche of wheat grains were mostly unidentified bacterial species. This finding highlights the lack of information that exists on the microorganisms that inhabit the endosphere of seeds and their role in plant growth, development, and health.

The fungal communities were dominated by a few phyla. *Ascomycota* was the most abundant phylum, with a relative abundance of 95.3% and 72.9% in the endophytic and epiphytic communities, respectively. The relative abundance of *Basidiomycota* was much higher in the epiphytic fungal community (26.1%) relative to fungal endophytic community (4.3%) (Figure 2A). At the genus level, OTUs of the fungal endophytes were assigned to *Alternaria* (76.8%), *Russula* (3.4%), *Stemphylium* (11.8%), *Aspergillus* (1.9%), *Epicoccum* (1.0%), *Cladosporium* (0.9%), *Pyrenophora* (0.5%), *Mycosphaerella* (0.4%), *Paradendryphiella* (0.4%), *Sporobolomyces* (0.1%) and *Filobasidium* (0.1%), and epiphytes annotated to *Alternaria* (50.3%), *Sporobolomyces* (9.5%), *Filobasidium* (8.8%), *Vishniacozyma* (4.2%), *Mycosphaerella* (3.6%), *Stemphylium* (3.5%), *Cladosporium* (10.5%), *Epicoccum* (1.4%), *Dioszegia* (1.1%), *Aspergillus* (0.8%), *Aureobasidium* (0.7%), *Paradendryphiella* (0.2%), *Russula* (0.1%) and *Pyrenophora* (0.1%) (Figure 2B).

The Shannon α-diversity was higher in the epiphytic community of both bacteria and fungi than it was in the endophytic community (Figure 3). A two-sample t-test based on the Shannon index revealed a significant difference between the structure of the endophytic and epiphytic bacterial (*p* = 0.001) and fungal (*p* = 0.001) community (Appendix A). PCoA was used to measure the dissimilarity of the epiphytic and endophytic microbial communities of wheat grains (Figure 4). The structure of the epiphytic and endophytic bacterial communities were clearly different as evidenced by their evident separation in the PCoA plot, and the same was true for the structure of the epiphytic and endophytic fungal communities. The ANOSIM test (nonparametric 999 permutations) of variance analysis also indicated that the composition of epiphytic and endophytic bacterial and fungal communities was significantly different (*R*^2^ = 0.89, *p* = 0.001 and *R*^2^ = 0.85, *p* = 0.001), respectively.

Endophytic microbial diversity was significantly lower than that of the epiphytic microbiota. This can be attributed to the need for seed endophytes to pass through the plant ecological, and physiological sieve to become internalized and tolerate harsh seed conditions, including a high level of desiccation.

### 3.2. Microbe Isolation from Stored Grains and the Antagonistic Activity of Selected Isolates against Mycotoxigenic Fungal Pathogens

A total of 159 bacterial isolates were cultured from wheat grains and identified by 16S rRNA sequencing; 118 of which were isolated from the grain surface as epiphytes and 41 as endophytes from the inner tissues of the grains. *Bacillus* spp. were dominant among cultured bacterial epiphytes (53.4%), followed by *Pseudomonas* spp. (10%), *Pantoea* spp. (6.8%), *Paenibacillus* spp. (5%), and *Staphylococcus* spp. and *Enterobacter* spp. (3.4% for each genus) (Figure 5A). Diversity among endophytes was lower than for epiphytes at the taxonomic level of genus. Similarly to bacterial epiphytes, however, most endophytes isolated from stored wheat grains belonged to the genera *Bacillus* (46.3%) and *Pseudomonas* (14.6%) (Figure 5A). These observations are consistent with previous studies of seed-associated endophytic bacteria, where *Bacillus* and *Pseudomonas* were the genera most frequently found in plant seeds [37]. In a study on barley seed endophytes, most endophytic bacteria were assigned to the same genera, with *Bacillus* and *Pseudomonas* being retrieved from the seeds placed under selective pressure for nitrogen-fixing microorganisms [38].

Among mold and yeast strains isolated from stored grains, 104 epiphytes and 37 endophytes were identified by ITS and D1/D2 region sequencing. The epiphytic fungal communities were the richest and most abundant, while richness and diversity in the endophytes was much lower. Most of the epiphytic and endophytic filamentous fungi were identified as *Alternaria*, *Aspergillus*, *Fusarium,* and *Penicillium* species. *Filobasidium*, *Cryptococcus*, *Rhodotorula,* and *Candida* species dominated both epiphytic and endophytic yeast cultures isolated from grains (Figure 5B). In general, the composition of the bacterial and fungal taxa isolated from wheat grains in our study were in agreement with previous reports [12,13,36,37,39]. The culturing results were also consistent with the results obtained by high-throughput sequencing in the present study. Bacterial communities assigned to *Pantoea*, *Pseudomonas*, *Bacillus*, *Enterobacter*, and *Paenibacillus* spp., and fungal microbiota comprising *Alternaria*, *Aspergillus*, *Fusarium,* and *Filobasidium* spp. were identified as the common genera among the members of the shared epiphytic and endophytic microbiota of stored wheat seeds.

In vitro screening of microbial isolates in dual-culture assays with *Aspergillus flavus, Fusarium proliferatum,* and *Alternaria infectoria* revealed only four *Bacillus* isolates that exhibited considerable inhibitory activity against the tested mycotoxigenic fungi (Appendix A). These were two endophytes (*B. amyloliquefaciens* strain EnB28 and *B. amyloliquefaciens* strain EnB29) and two epiphytes (*B. licheniformis* strain EpV5 and *B. subtilis* strain EpV29). The isolates appeared to secrete antifungal substances that were capable of inhibiting fungal growth (Appendix A). The mean values of pathogen growth inhibition fluctuated between 23 and 40% against *Aspergillus flavus*, 23 and 41% against *F. proliferatum*, and 38 and 54% against *Alternaria infectoria*. In contrast, a volatile organic compound (VOC) assay revealed that none of the isolated exhibited inhibitory activity against any of the tested fungi. These results conflict with previous studies in which VOC inhibitory activity by *Bacillus* strains against *Aspergillus*, *Fusarium*, and *Alternaria* species has been reported [40,41,42]. 

To validate our in vitro results on the biocontrol potential of *Bacillus* isolates, we designed a simple bioassay suitable for analyzing antagonistic activity of the selected *Bacillus* strains against mycotoxigenic fungi in stored grains. All of the bacterial isolates tested significantly reduced fungal growth in grains compared to untreated controls (*p* < 0.05, Figure 6). The tested *Bacillus* species exhibited the strongest antagonistic activity against *F. proliferatum*, resulting in an up to a 3.59 log_10_ CFU/g reduction in the fungal population compared to control samples. Additionally, up to a 2.45 and 1.76 log_10_ CFU/g decrease was recorded for *Aspergillus flavus* and *Alternaria infectoria*, respectively. *Bacillus* spp., such as *B. amyloliquefaciens* and *B. subtilis*, have been previously identified and characterized as antagonistic toward *Fusarium* spp. on wheat [14,43,44,45,46,47,48,49]. Few studies, however, have been conducted on the antagonistic potential of *Bacillus* isolates against *Aspergillus* and/or *Alternaria* pathogens on wheat [50]. *Bacillus* strains are well known for their ability to produce a wide range of potent antimicrobial lipopeptides, including iturins, surfactins, fengycins, polymyxins, and others, that have broad effects on phytopathogens [51,52,53]. Although they were not characterized in this study, some of these compounds could be responsible for the bacterial inhibitory activity against the tested fungal pathogens. In addition to lipopeptides, other antimicrobial compounds produced by *Bacillus* strains, such as polyketides and lithium enzymes [54], may also be involved in the antifungal activity observed both in vitro and in vivo. Determining the source of the antimicrobial activity of the *Bacillus* isolates against the tested fungal pathogens will require further study.

In sharp contrast to the bacterial strains, yeast isolates obtained from wheat seeds exhibited no inhibition of mycotoxigenic fungal growth in vitro in the dual-culture assay. Further evaluation of the yeast strains by screening for activity on colonized wheat grains revealed four species that exhibit antagonistic properties against the tested fungal pathogens. The yeast isolates were *Naganishia albidosimilis* (strain D1), *Naganishia albida* (strain D34), *Cryptococcus albidus* (strain D37), and *Rhodotorula glutinis* (strain TY1). As shown in Figure 7, the growth of *F. proliferatum* on wheat seeds was markedly limited when the grains were pretreated with yeast cell suspensions. Among them, *R. glutinis* TY1 displayed a strong ability to protect the stored grains against *F. proliferatum* by reducing the population of the fungal load by 3.11 log_10_ CFU/g. This strain also displayed the strongest antagonistic activity against *Aspergillus flavus* with a reduction of 1.33 log_10_ CFU/g in the fungal load compared to the untreated control. All four yeast isolates, however, had a relatively weak effect on *Alternaria infectoria*.

The differential efficacy observed between the in vitro and in vivo screening of yeast isolates could be explained by the different abilities of the potential biocontrol strains to produce secondary metabolites in a growth medium a vs. on plant material [12,55]. Therefore, determination of an inhibition zone by dual-culture assay, often considered a preliminary test for selecting antagonistic strains, can be misleading [12].

### 3.3. Mycotoxin-Degrading Activity of the Yeast Rhodotorula glutinis Strain TY1

In vitro growth kinetics of *R. glutinis* TY1 in the absence and presence of AFB_1_ were similar and characterized by exponential growth until the third day of incubation (Figure 8). Subsequent HPLC analysis indicated that the recovery of AFB_1_ was lower when *R. glutinis* strain TY1 was incubated in PDB medium amended with 250 ng/mL AFB_1_. Reductions of 47, 61, and 75% in AFB_1_ recovery were observed on days 1, 2, and 3 of incubation, respectively, compared to the control (Figure 8A). In stored grain samples artificially contaminated with AFB_1_, a decrease in mycotoxin levels was recorded throughout the time course of the experiment for TY1-treated seeds compared to untreated controls. The presence of the yeast strain significantly lowered mycotoxin levels in grain samples by 26.5, 51, and 65% on days 1, 2, and 3 of the experiment, respectively, compared to untreated controls (Figure 8B).

AFB_1_ is a mycotoxin that presents an economic challenge to a wide range of agricultural products and food industries worldwide, and a health hazard to consumers. The reduction of aflatoxin contamination in commodities through the application of antagonistic microorganisms has long been known, and is still an active field of research [56]. Castoria et al. [34] showed that *R. glutinis* can decrease patulin in vitro and in apple fruit infected by *Penicillium expansum*. Their data indicated that *R. glutinis* yeast cells metabolize patulin and/or negatively affect its accumulation or synthesis. Other studies have reported that the biocontrol yeast *Sporobolomyces* sp. degrades patulin in vitro under aerobic conditions and converts it into less toxic breakdown products [57,58]. Based on the results obtained in the current study, it appears that *R. glutinis* can metabolize and/or degrade AFB_1_ in vitro and in stored wheat grains. The mechanism responsible for this activity, however, still needs to be confirmed. Moreover, characterization of the final breakdown products of AFB_1_ and their toxicological properties also need to be investigated, to ensure the safety of using this yeast isolate as a postharvest application on stored grains. 

## 4. Conclusions

In summary, we identified the composition of stored wheat grain-associated epiphytic and endophytic microbiota using high-throughput sequencing of amplicons. Large numbers of microorganisms forming epiphytic and endophytic communities often possess beneficial characteristics, including antagonistic activity against fungal pathogens. Indeed, we found that stored wheat seeds harbor bacterial and yeast communities with antagonistic potential and biodegradation ability against mycotoxigenic fungi and their respective toxins. Application of such microbes, which are adapted to their host plant, may represent an efficient fungal disease control strategy for stored grains, as well as other agricultural commodities. A better understanding of the antagonistic mechanism of these microorganisms, and the role they play in the microbiome of seeds, may assist in the development of novel antifungal biocontrol approaches to replace traditionally used synthetic fungicides. 

## Figures and Tables

**Figure 1 jof-07-00781-f001:**
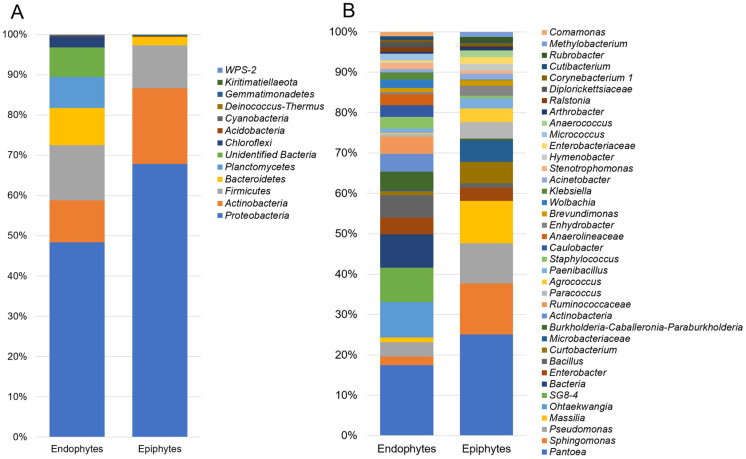
Bar charts showing the relative abundance of most dominant bacterial phyla (**A**) and genera (**B**) among endophytes and epiphytes detected in stored wheat grain samples using high-throughput sequencing technology.

**Figure 2 jof-07-00781-f002:**
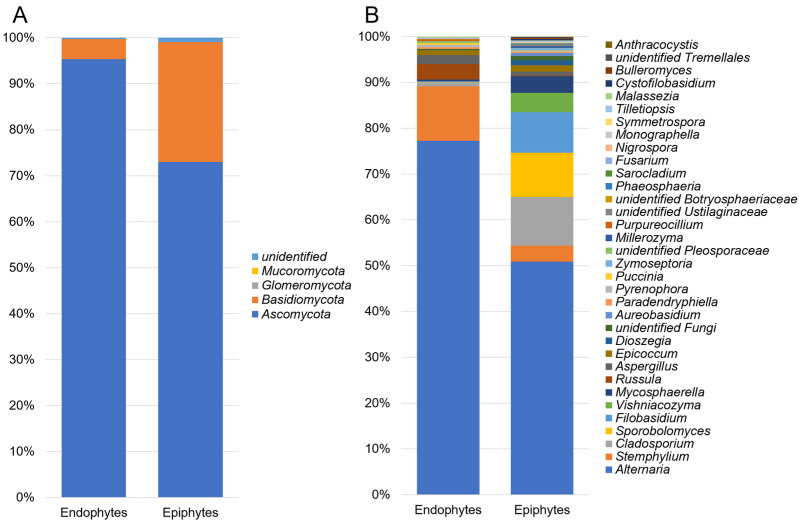
Relative abundance of fungal phyla (**A**) and genera (**B**) among endophytes and epiphytes detected in grain samples using high-throughput sequencing technology.

**Figure 3 jof-07-00781-f003:**
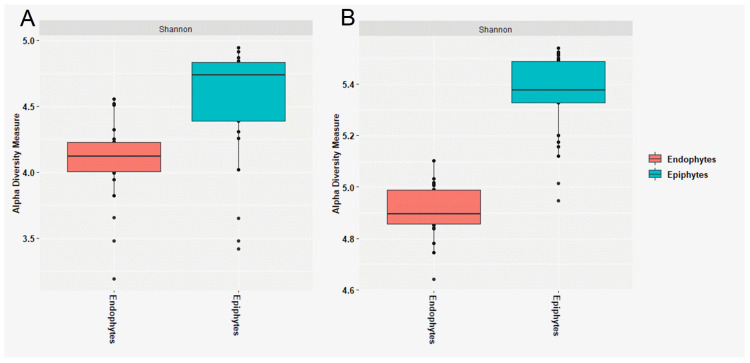
Alpha diversity analysis (based on Shannon diversity index) of the bacterial (**A**) and fungal (**B**) epiphytes and endophytes in stored wheat grain samples. The centered square represents the mean, black line inside the box represents the median, and circles indicate outliers.

**Figure 4 jof-07-00781-f004:**
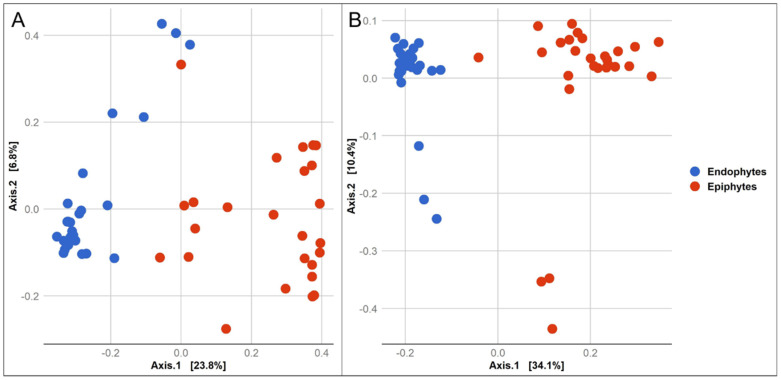
Principal coordinate analysis (PCoA) based on the beta diversity Bray Curtis dissimilarity metrics, showing the distance in the wheat grain-associated bacterial (**A**) and fungal (**B)** communities between endophytic and epiphytic microbes.

**Figure 5 jof-07-00781-f005:**
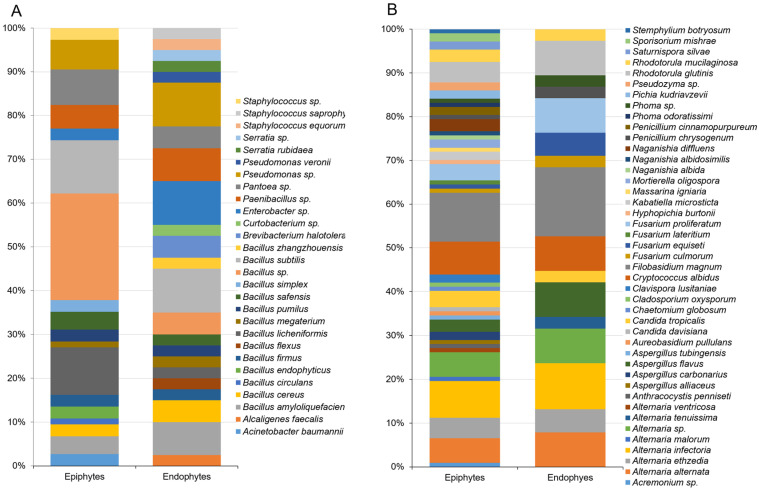
Relative abundance of bacterial (**A**), and fungal (**B**) species isolated by plating from epiphytic and endophytic microflora of stored wheat grains. The isolates were identified by sequencing the 16S rRNA gene in bacteria, ITS region and D1/D2 domain of large-subunit rDNA in filamentous fungi and yeasts, respectively. The sequences were determined via BLAST matches to the NCBI database.

**Figure 6 jof-07-00781-f006:**
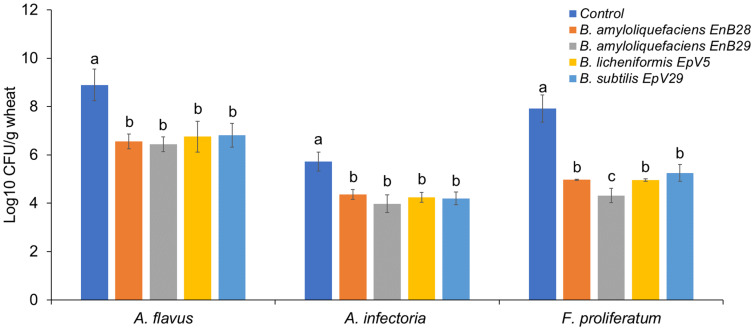
Antagonistic activity of four selected bacterial isolates against three mycotoxigenic fungal pathogens in stored wheat grains. Data are means of at least three independent repetitions ± standard error. One-way ANOVA differences were considered significant when *p* < 0.05. Different letters above the error bars indicate statistically significant differences among treatments in each group as determined based on Duncan’s multiple range test.

**Figure 7 jof-07-00781-f007:**
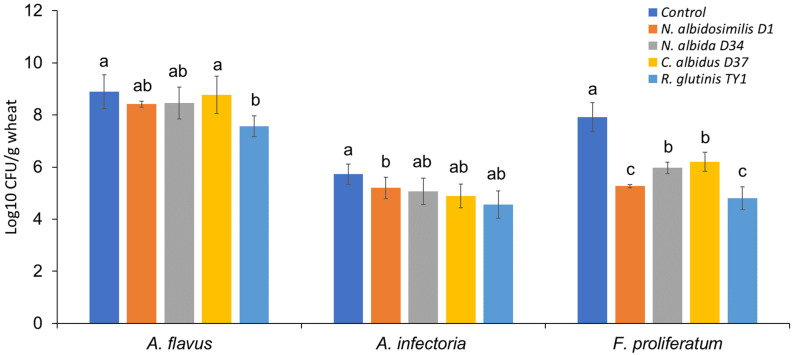
Antagonistic activity of four selected yeast isolates against three mycotoxigenic fungi in stored wheat grains. Error bars represent standard error of three independent biological replicates. Different letters above the error bars indicate statistically significant differences among treatments in each group at *p* < 0.05, as determined based on Duncan’s multiple range test.

**Figure 8 jof-07-00781-f008:**
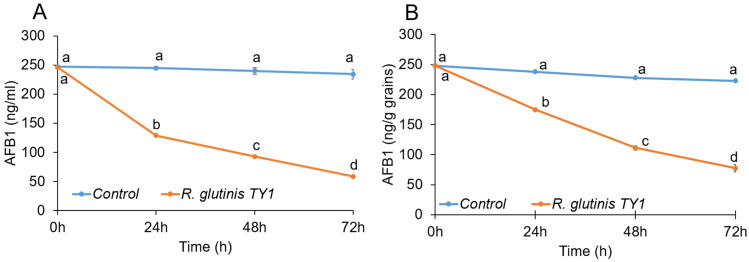
Effect of *R. glutinis* TY1 strain on AFB1 degradation in PDB medium (**A**) and wheat grains (**B**). The time course of AFB1 degradation was monitored by HPLC. Error bars represent the standard error of the mean (SEM) across three independent replicates. One-way ANCOVA was conducted to determine a statistically significant difference between treatment and control on mycotoxin concentration controlling for time (covariance). Time significantly differed in its effect on treatment and control (*p* < 0.0001). Testing for each time point using ANOVA is an evident that only time zero does not differ between control and treatment. Different letters above the error bars indicate statistically significant differences at *p* < 0.05, as determined using the Duncan’s multiple range test.

## Data Availability

The raw sequence files supporting the findings of this article are available in the NCBI Sequence Read Archive (SRA) under the BioProject ID PRJNA756939.

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
