# Peer review of "Analysis of Stored Wheat Grain-Associated Microbiota Reveals Biocontrol Activity among Microorganisms against Mycotoxigenic Fungi"

_jof, 2021, doi:10.3390/jof7090781_

Round 1

Reviewer 1 Report

In my opinion this is a very interesting and important article dealing with 2 very important problems ,1) mycotoxin contamination of one of the most important food stuffs and b) the use  syntehic persistent toxic fungisides for control of mycotoxigenic fungi. The article offers a replacement of syntetic fungisides by antagoistic  microbes and presents interesting and promising results.

I have two minor comments:

Figure 5: The logic of the different colours represneting diffents microbes took me a while to understand. May be a explaining sentence in the figure legend would make it easier for a more stupid reader.

Figure 6: I did not understand what the letters a-c meant. In my opinion this needs an explanation comprehensive for the reader.

Just optional: a potograph of an example of  illustating the measured in vivo and/or  in vitro antagonism would be interesting to see.

Author Response

We thank the reviewer for his comments. Below is our response to the comments.

Figure 5: The logic of the different colours represneting diffents microbes took me a while to understand. May be a explaining sentence in the figure legend would make it easier for a more stupid reader.

The figure demonstrates the relative abundance of bacterial and fungal species isolated by plating from epiphytic and endophytic microflora of stored wheat grains. Yes, the figure is a quite busy, but, unfortunately, we did not find a better explanation for the data. The figure could be enlarged, then any reader could easily refer each color with species name to specific part in the figure with the same color to see its part in microbiome.

Figure 6: I did not understand what the letters a-c meant. In my opinion this needs an explanation comprehensive for the reader.

Figures 6 and 7 were reorganized; as was mentioned in the legends, the letters (a, b, c) above the error bars indicate statistically significant differences among treatments by bacterial (Figure 6) or yeast (Figure 7) isolates compared to untreated controls.

Just optional: a potograph of an example of  illustating the measured in vivo and/or  in vitro antagonism would be interesting to see.

According to the reviewer's suggestion, Figure S2 is added to the Supplementary Material showing in vitro antagonism of one of the representative bacteria Bacillus amyloliquefaciens (strain EnB28) against the mycotoxigenic fungal pathogens.

Reviewer 2 Report

These are my main comments on the MS (jof-1375842) entitled : “Analysis of Stored Wheat Grain-Associated Microbiota Reveals Biocontrol Activity among Microorganisms against Mycotoxigenic Fungi”.

It is an interesting dealing with the inhibition effect of some grain microbiota against some harmful fungi. This basic knowledge will help us to enhance the use of beneficial microbes in storage facilities.

I identified serious gaps and errors in the methodology and presentation of results that should be corrected before publication. My proposal is to accept it for publication in your journal after major revision.

Comments

Abstract. There are no numerical data. I think that some critical values should be presented to highlight the effect of these antagonistic microorganisms.

Lines 173-177. Authors should add a sub-section describing the source, collection, identification, and culture technique of these three fungal species (Aspergillus flavus, Fusarium proliferatum and Alternaria infectoria) and the methodology they adopted to make the conidial suspension.

Lines 187-188. How was the inhibition growth measured? There is no description here. How was the fungal growth measured? With a ruler, with a software like Image J etc?

Line 189. “At least three independent repetitions…”. Do you mean that there were more replicates in some treatments? I think that there were only 3, not at least 3. Three replications are not enough for dual culture tests. There should be at least 5-6.

Line 259. “…one-way analysis of variance …. to determine treatment effects and interactions….”. You cannot determine interaction effect with simple ANOVA. You need multiple – way ANOVA.

Statistical Analysis. Please state the statistical package that was used with the relevant reference (eg. SPSS, GenStat).

Results are quite many to merge with Discussion. I suggest dividing them.

Figs 6,7 & S1. These figures are very hard to study because of wrong presentation setup. I suspect that ANOVA letters present differences among the same fungus (same color bars). In such case the same fungus bars should be put together. Also, in Fig S1 it seems that letters are mistakenly used in A.infectoria bars. Please check it.

Lines 288, 292, 330, 331, 394 etc. When significant differences among treatments are mentioned in the text, the p < a expression is not enough. All statistical parameters should be presented (df, F value and the exact p value). Authors have these values from the statistical analysis. They should add them in the text to show the level of significance in differences among treatments.

Line 481-482. “….we found that stored wheat seeds harbor bacterial and yeast antagonists with strong inhibition potential….”. In most cases the inhibition growth was 30-40%. In natural conditions this would be even lower. I do not think that this can be characterized as “strong inhibition potential”. Authors should discuss their results more carefully.

Author Response

We would like to thank the reviewer for the useful comments. Here is our response to the comments.

Abstract. There are no numerical data. I think that some critical values should be presented to highlight the effect of these antagonistic microorganisms.

We added the numerical data into the abstract according to the reviewer suggestion.

Lines 173-177. Authors should add a sub-section describing the source, collection, identification, and culture technique of these three fungal species (Aspergillus flavus, Fusarium proliferatum and Alternaria infectoria) and the methodology they adopted to make the conidial suspension.

All required information regarding fungal species used in the study was added in a sub-section 2.5 "In Vitro Antagonism against Mycotoxigenic Fungi" (Lines 175-183 in the revised version).

Lines 187-188. How was the inhibition growth measured? There is no description here. How was the fungal growth measured? With a ruler, with a software like Image J etc?

Fungal growth inhibition was measured with a ruler, the sentence was corrected (Line 198 in the revised version).

Line 189. “At least three independent repetitions…”. Do you mean that there were more replicates in some treatments? I think that there were only 3, not at least 3. Three replications are not enough for dual culture tests. There should be at least 5-6.

Five dual-culture independent tests were performed between yeasts and mycotoxigenic fungi; in these experiments we did not see fungal growth inhibition by yeast isolates. In contrast, in all three biological experiments of dual-culture assays between bacterial and fungal isolates (with 2-3 technical replications in each experiment) significant inhibition of fungal growth with high reproducibility of the results were observed; therefore, in our opinion there was no need to perform further experiments.

Line 259. “…one-way analysis of variance …. to determine treatment effects and interactions….”. You cannot determine interaction effect with simple ANOVA. You need multiple – way ANOVA.

We appreciate reviewer's comment. Indeed, we performed one way ANOVA to determine only treatment effects; we did not determine interaction effects - the sentence was corrected accordingly.

Statistical Analysis. Please state the statistical package that was used with the relevant reference (eg. SPSS, GenStat).

We added the statistical package (SPSS) in "Statistical Analysis" sub-section. 

Results are quite many to merge with Discussion. I suggest dividing them.

We prefer to keep a unified chapter of results and discussion, because dividing the results and discussion at this stage will cause unwanted changes in the article and many confusions.

Figs 6,7 & S1. These figures are very hard to study because of wrong presentation setup. I suspect that ANOVA letters present differences among the same fungus (same color bars). In such case the same fungus bars should be put together. Also, in Fig S1 it seems that letters are mistakenly used in A.infectoria bars. Please check it.

Figures 6, 7 and S1 were reorganized and the legends were corrected accordingly.

Lines 288, 292, 330, 331, 394 etc. When significant differences among treatments are mentioned in the text, the p < a expression is not enough. All statistical parameters should be presented (df, F value and the exact p value). Authors have these values from the statistical analysis. They should add them in the text to show the level of significance in differences among treatments.

In the lines mentioned by the reviewer, we refer to Shannon diversity and differential abundance analysis. Both tests do not produce F values nor have degrees of freedom because the first is t-test and the second is Kruskal-Wallis. Also, the reason why we had “P < 0” is because we were referencing to a list of taxa, each of which had different P values. Nevertheless, in the Supplementary Material we attached the results of the differential abundance taxa (based on Kruskal-Wallis, Tables S2, S3) and alpha diversity (Shannon based on t-test, Table S4), and mentioned them accordingly in the text.

Line 481-482. “….we found that stored wheat seeds harbor bacterial and yeast antagonists with strong inhibition potential….”. In most cases the inhibition growth was 30-40%. In natural conditions this would be even lower. I do not think that this can be characterized as “strong inhibition potential”. Authors should discuss their results more carefully.

The sentence was rephrased according to the reviewer's suggestion. 

Round 2

Reviewer 2 Report

Authors have responded adequately to many of my suggestions. The MS can be accepted for publicaiton.